# The Inhibition of the Membrane-Bound Transcription Factor Site-1 Protease (MBTP1) Alleviates the p.Phe508del-Cystic Fibrosis Transmembrane Conductance Regulator (CFTR) Defects in Cystic Fibrosis Cells

**DOI:** 10.3390/cells13020185

**Published:** 2024-01-18

**Authors:** Raphaël Santinelli, Nathalie Benz, Julie Guellec, Fabien Quinquis, Ervin Kocas, Johan Thomas, Tristan Montier, Chandran Ka, Emilie Luczka-Majérus, Edouard Sage, Claude Férec, Christelle Coraux, Pascal Trouvé

**Affiliations:** 1Univ Brest, Inserm, EFS, UMR 1078, 22 Avenue Camille Desmoulins, F-29200 Brest, France; raphael.santinelli@univ-brest.fr (R.S.); nathalie.benz@inserm.fr (N.B.); julie.guellec@kalsiom.fr (J.G.); fabien.quinquis@gmail.com (F.Q.); ervinkocas@gmail.com (E.K.); johanthomas@free.fr (J.T.); tristan.montier@univ-brest.fr (T.M.); chandran.ka@univ-brest.fr (C.K.); claude.ferec@univ-brest.fr (C.F.); 2Inserm UMR-S 1250, University of Reims Champagne-Ardenne (URCA), SFR Cap-Santé, F-51100 Reims, France; emilie.luczka1@univ-reims.fr (E.L.-M.); christelle.coraux@univ-reims.fr (C.C.); 3Université Paris-Saclay, INRAE, UVSQ, VIM, F-78350 Jouy-en-Josas, France; e.sage@hopital-foch.com

**Keywords:** Cystic Fibrosis, p.Phe508del-CFTR, MBTP1, ATF6

## Abstract

Cystic Fibrosis (CF) is present due to mutations in the Cystic Fibrosis Transmembrane Conductance Regulator (CFTR) gene, the most frequent variant being p.phe508del. The CFTR protein is a chloride (Cl-) channel which is defective and almost absent of cell membranes when the p.Phe508del mutation is present. The p.Phe508del-CFTR protein is retained in the endoplasmic reticulum (ER) and together with inflammation and infection triggers the Unfolded Protein Response (UPR). During the UPR, the Activating Transcription Factor 6 (ATF6) is activated with cleavage and then decreases the expression of p.Phe508del-CFTR. We have previously shown that the inhibition of the activation of ATF6 alleviates the p.Phe508del-CFTR defects in cells overexpressing the mutated protein. In the present paper, our aim was to inhibit the cleavage of ATF6, and thus its activation in a human bronchial cell line with endogenous p.Phe508del-CFTR expression and in bronchial cells from patients, to be more relevant to CF. This was achieved by inhibiting the protease MBTP1 which is responsible for the cleavage of ATF6. We show here that this inhibition leads to increased mRNA and p.Phe508del-CFTR expression and, consequently, to increased Cl-efflux. We also explain the mechanisms linked to these increases with the modulation of genes when MBTP1 is inhibited. Indeed, RT-qPCR assays show that genes such as HSPA1B, CEBPB, VIMP, PFND2, MAPK8, XBP1, INSIG1, and CALR are modulated. In conclusion, we show that the inhibition of MBTP1 has a beneficial effect in relevant models to CF and that this is due to the modulation of genes involved in the disease.

## 1. Introduction

Cystic Fibrosis (CF) is the most common lethal autosomal recessive disease in the European population. It is mainly characterized by pulmonary disorders and is due to a large panel of mutations in the *Cystic Fibrosis Transmembrane Conductance Regulator* (CFTR) gene affecting the CFTR protein synthesis and activity [1,2]. The CFTR protein is an N-glycosylated transmembrane protein, a member of the ABC transporter family [3,4]. An electrophoretic analysis shows that CFTR exists in three different forms of 130, 135, and 150 kDa referred to as Band A, B, and C, respectively [5]. These bands represent the different glycosylated forms of CFTR. Band A is the non-glycosylated form, Band B is the core-glycosylated CFTR, and Band C is the mature form of CFTR with complex glycosylation [4]. The glycosylation state of CFTR is thus representative of its maturation and is an important marker of its processing and function as a cAMP-activated, phosphorylation-regulated Cl-channel, responsible for the transport of chloride (Cl-) and bicarbonate anions [2,3,6]. Linking with the Epithelial Sodium Channel (ENaC), it maintains proper hydration of the mucus [7].

The most frequent mutation in CF is a phenylalanine deletion at position 508 of the polypeptide sequence (p.Phe508del) [5,8]. This deletion alters the folding of the protein (p.Phe508del-CFTR) which is retained within the Endoplasmic Reticulum (ER), which cannot be validated with the quality control of the ER and is rapidly degraded by the Endoplasmic Reticulum Associated Degradation (ERAD) [8,9]. p.Phe508del-CFTR remains in a core-glycosylated form, and only a negligible amount reaches the plasma membrane of the cells [10,11]. The consequence in the lung, together with the lost inhibition of ENaC, is the presence of a more viscous mucus that diminishes the mucociliary clearance and enables pathogens to develop, inducing chronic infections and inflammation [12,13]. The subsequent altered lung function is the main cause of morbidity and mortality in CF.

Inflammation, infection, and the retention of misfolded proteins within the ER are cellular stressors triggering the Unfolded Protein Response (UPR) [14,15]. The UPR is a normal physiological recovery process aimed to regulate the protein load in the ER and to alleviate the cellular stress. The 78 kDa glucose-related protein/Binding-immunoglobulin protein (Grp78/BiP) activates the UPR through its binding to the unfolded protein [16,17]. This binding leads to its dissociation from three main effectors which are Inositol-Requiring Enzyme 1α (IRE1α) [18], Protein kinase R (PKR)-like Endoplasmic Reticulum Kinase (PERK) [19] and Activating Transcription Factor 6 (ATF6) [20,21]. These effectors then activate the transcription of genes encoding molecular chaperones, folding catalysts, and proteins involved in the ERAD. They also decrease the global synthesis of proteins in order to avoid an overload of the ER [16,22]. ATF6 (90 kDa) allows the transcription of chaperones, of proteins related to misfolded protein degradation, and of the cholesterol metabolism but it also decreases the expression of some genes such as *CFTR* [23,24]. When GRP78 dissociates, ATF6 migrates and anchors in the membranes of the Golgi apparatus where it is successively cleaved with MBTP1 and two that cleave the lumen section of ATF6 and the juxta-membrane region at the cytosol side, respectively [20,25,26]. The cleaved form of ATF6 (50 kDa) is subsequently released into the cytosol and migrates to the nucleus where it acts as a transcription factor [24,27,28]. Whereas ATF6 negatively regulates the expression of CFTR, we showed that its inhibition using siRNA in transfected A549 cells expressing p.Phe508del-CFTR restores the Cl-flux, making ATF6 a potential therapeutic target for CF [29,30,31,32]. More recently, we inhibited the cleavage of ATF6 in CFBE41o-cells transduced to overexpress p.Phe508del-CFTR and found that this inhibition also alleviates the defects due to the mutation in these cells [33].

In the present study, we inhibited MBTP1 and thus the cleavage and activation of ATF6 to alleviate p.Phe508del-CFTR defects, in a non-transduced Human Bronchial Cell line and in bronchial epithelia from patients, to be as close as possible to the physiopathology of CF in a relevant cell model. Indeed, these non-transduced CF cells freed us from possible biases due to the CFTR transduction, with respect to UPR triggering. Furthermore, we assessed the effects of the inhibition of MBTP1 in human bronchial epithelia from CF patients, which has never been performed before. We found that this inhibition increases the p.Phe508del-CFTR’s expression and the Cl-efflux, due to an augmented presence of the core-glycosylated form of CFTR in the membranes of the cells. In order to explain how the inhibition of MBTP1 acts upon Cl-efflux, we performed RT-qPCR-Arrays and highlighted that the HSPA1B, CEBPB, VIMP and Calreticulin (CALR) genes are the main genes involved in the alleviation of the p.Phe508del-CFTR’s defects in our models.

In conclusion, we showed that the protease MBTP1 is a potential target against CF.

## 2. Material and Methods

### 2.1. Cell Culture and Protein Extraction

The native CFBE41o-cells and the transduced CFBE41o-cell lines (CFBE41o-/corrected (corr) and CFBE41o-/F508del) were cultured, as previously described [34]. These cells are human bronchial epithelial cells, derived from a CF patient homozygous for the p.Phe508del mutation and immortalized with an SV40 plasmid (pSVori-) [34,35]. Cells were obtained from the Cystic Fibrosis Foundation (Bethesda, MD, USA). In some experiments, cells were treated with PF-429242 (PF) that specifically inhibits MBTP1 [36] (10 µM for 48 h; SML0667, Sigma-Aldrich, Saint-Quentin Fallavier, France); and/or with thapsigargin (0.5 µM for 4 h; #10798352, ThermoFischer, Saint Rémy les Chevreuses, France) to induce UPR; and/or with VX-809 (3 µM for 48 h; S1565, Selleckchem, Cologne, Germany).

For protein extraction, cells were washed with cold PBS ×1 and lyzed in RIPA buffer (25 mM Tris, 150 mM NaCl, 1% Triton ×-100, 1% Na-Deoxycholate, 0.1% SDS, 10 mM iodoacetamide, 100 mM PMSF; pH = 7.5) in the presence of the Complete Protease Inhibitor Cocktail (PI; 40 µL/mL, Complete tablets EDTA free, Roche, Mannheim, Germany).

For Ussing chamber experiments, human-airway epithelial cells were obtained from the department of thoracic surgery and lung transplantation of Foch hospital (Suresnes, France).

### 2.2. Nuclear Extract Preparation

Cells were washed twice with cold PBS 1×, scrapped in the presence of PI and centrifuged (5 min, 1000× *g*, 4 °C). The pellets were suspended in Lysis buffer (0.3 M Sucrose, 1 mM Sodium azide, Hepes 20 mM; pH = 7.2) in the presence of PI before 10 min incubation on ice. Lysates were centrifuged (10 min, 2000× *g*, 4 °C), and pellets were suspended in RIPA buffer with PI. Lysates were centrifuged (10 min, 800× *g*, 4 °C), and pellets were suspended in NaI8 buffer (0.1 M NaCl, 8× Sucrose, 30 mM Imidazole) with PI.

### 2.3. Membrane Extract Preparation

Cells were washed twice with cold PBS 1×, scrapped in the presence of PI, and centrifuged (5 min, 1000× *g*, 4 °C). The pellets were suspended in Lysis buffer (1 mM EDTA, Hepes 10 mM; pH = 7.2) with PI before 10 min of incubation on ice. Mechanical lysis was performed using Dounce homogenizer (VWR International, Rosny-sous-Bois, France) eight times back and forth. Sucrose buffer was added (500 mM Sucrose, Hepes 10 mM; pH = 7.2) and a further eight times of lysis back and forth was performed. Lysates were centrifuged (10 min, 6000× *g*, 4 °C), and the supernatant was ultracentrifuged (30 min, 100,000× *g*, 4 °C). Pellets were suspended in NaI8 buffer with PI.

### 2.4. Immunoprecipitation

After protein extraction, lysates (700 µg protein) were subjected to pre-clearing steps with magnetic beads (Immunoprecipitation Kit—Dynabeads™ Protein G, Invitrogen, Carlsbad, CA, USA) for 2 h at 4 °C. The supernatant was incubated with 2 µg anti-CFTR antibody (24-1 clone, Novus Biologicals, Littleton, CO, USA) overnight at 4 °C. The antibody–antigen complex was incubated with the magnetic beads for 3 h at 4 °C, then washed three times. In order to load the complex on SDS-PAGE, 2× Laemmli Sample Buffer plus β-mercaptoethanol was added on the bead–antibody–antigen complexes before heating at 75 °C for 10 min.

### 2.5. Cell Viability in the Presence of the MBTP1 Inhibitor

The viability of the cells after treatment was measured using a colorimetric MTT (3-(4,5-dimethylthiazol-2-yl)-2,5-Diphenyltetrazolium bromide) test (MTT kit, Millipore, Burlington, MA, USA), according to the manufacturer’s instructions. In brief, CFBE41o-cells were cultured in a 96-well plate and treated with PF. After the treatment, WST-8 (2-(2-methoxy-4-nitrophenyl)-3-(4-nitrophenyl)-5-(2,4-disulfophenyl)-2H-tetrazolium, monosodium salt) was added to the wells and incubated for 4 h in a humidified incubator (37 °C, 5% CO_2_). WST-8 was reduced to an orange formazan product that was directly proportional to the number of living cells, which were detected by measuring the absorbance at 450 nm. The percentage of viability of the cells exposed to the MBTP1 inhibitor was determined through comparing their absorbance with that of the untreated cells (100% of viability).

### 2.6. Western Blotting

Cell lysates were centrifuged (15 min, 16,000× *g*, 4 °C), and protein concentrations were determined using the Lowry’s methodology. Samples were subjected to SDS/PAGE and transferred onto membranes which were then incubated with anti-GRP78 antibody (NBP1-06274, Novus Biologicals), anti-ATF6 antibody (MAB6762 clone 1-7, Abnova, Taipei, Taiwan), anti-SREBP2 (#7076, Cell Signaling, Limoges, France), anti-aldolase antibody (ab169544, Abcam, Paris, France), anti-Histone H1 antibody (sc-8030 AE-4, Santa Cruz Biotechnology, Heidelberg, Germany), anti-CFTR antibody (596, US Cystic Fibrosis Foundation), anti-CFTR antibody (24-1 clone, Novus Biologicals), anti-NaK-ATPase antibody (ab76020, Abcam), and anti-Actin-HRP (sc-47774, Santa Cruz Biotechnology). Densitometric measurements were performed using Image Studio Lite software (Version 5.2) and signals were normalized with the housekeeping genes.

### 2.7. Immunofluorescence

Cells were cultured in Nunc Lab-Tek II Chamber Slide System™. After 48 h, cells were permeabilized by paraformaldehyde 4% at room temperature. Nonspecific binding sites were saturated with BSA (3%, 2 h). Incubations were performed with the ATF6 primary antibodies (1:500; Merck-Millipore, 09-069, Burlington, MA, USA) and with the anti-PDI antibody (1:1000, Thermo Fisher Scientific, MA3-018,). The fluorophore-tagged secondary antibodies were Alexa Fluor 488^®^ (1:400; Jackson Immunosearch, Montlucon, France AB2313584) and Cyanine 3 (1:400; Jackson Immunosearch, AB2338680). Nuclear counterstain was conducted with Vectashield Antifade Mounting Medium with DAPI (Vector Laboratories, Les Ulis, France). Images were obtained with a Zeiss Imager M2 microscope and an Axiocam 503 (Carl Zeiss, Oberkochen, Germany).

### 2.8. RNA Extraction and Reverse Transcription-Polymerase Chain Reaction (RT-PCR)

Total RNA from untreated and treated cells were extracted using NucleoSpin RNA plus columns (Macherey-Nagel, Hoerdt, France), according to the manufacturer’s instructions. In total, 400 ng to 1 µg of total RNA was used for the cDNA synthesis. Reverse transcription was performed with the SuperScript^®^ II reverse transcriptase (Invitrogen). A PCR was performed with a HotStarMastermix (Qiagen, Hilden, Germany). Primers were as follows: for IRE1α, for forward, 5′-gaaaaggaatccctggatgg and, for reverse, 5′-tcagagggcgtctggagtc; for XBP1, for forward, 5′-cagcgcttggggatggatgc and, for reverse, 5′-gggcttggtatatatgtgg; for PERK, for forward, gactacatatggactcagtgc5′- and, for reverse, agatgtcctcccttcttac 5′-; for CHOP, for forward, aaatgggggtacctatgtttcac 5′- and, for reverse, cggtcaatcagagctcgg 5′-; and for Actin, for forward, 5′-gttgctatccaggctgtg and, for reverse, 5′-cactgtgttggctacag. The thermal cycling conditions were 95 °C for 15 min followed by 40 cycles at 94 °C for 30 s, 61 °C for 30 s, 72 °C for 1 min, and a final extension at 72 °C for 10 min. The housekeeping β-actin genes were used as a control. The PCR products were resolved using a 1% agarose gel, prestained with BET, and visualized under UV light. The intensities of the bands were determined with densitometry using the ImageJ software (1.54h version). β-actin normalization was performed. A statistical analysis was performed with at least three independent experiments.

The expression of p.Phe508del-CFTR, in the presence of and without PF, was analyzed using real-time PCR (LightCycler 480, Roche). The synthesized cDNA was mixed with 1× SYBR Green Master Mix (Qiagen) and 10 µM of either the specific CFTR’s primers (FW 5′-atgcccttcggcgatgtttt, reverse: 5′-tgattcttcccagtaagagaggc) or the β-Actin primers. The real-time PCR cycling conditions were as follows: PCR enzyme activation step at 95 °C for 15 min; 45 cycles of denaturation at 95 °C for 30 s; annealing at 57 °C for 30 s; and extension at 72 °C for 30 s. All conditions were normalized relative to the β-Actin control transcript. The results were analyzed using the 2^−ΔΔCt^ method.

### 2.9. Patch Clamp

Patch-clamp experiments were performed with an automatic electrophysiology workstation (Port-a-Patch, Nanion Technologies GmbH, Hambourg, Germany) coupled to an external amplifier unit HEKA EPC-10 [37,38]. Whole-cell recordings were performed with treated or non-treated cells with the MBTP1 inhibitor. All measurements were obtained at room temperature. The voltage clamp protocol was carried out between −80 and +80 mV (10 mV per steps) with a holding membrane potential of −80 mV. The following buffers were used to suspend the cells: 140 mM NaCl, 2 mM CaCl2, 1 mM MgCl2, 10 mM Hepes (pH 7.4), 5 mM D-glucose monohydrate, 298 mOsm. The internal buffers were 50 mM CsCl, 10 mM NaCl, 60 mM Cs-Fluoride, 20 mM EGTA, 10 mM Hepes/CsOH, 5 mM Mg-ATP; pH 7.2; 285 mOsmol. CFTR’s activators (forskolin, 10 µM and genistein, 30 µM; Sigma Aldrich, Saint-Quentin Fallavier, France) and inhibitor (CFTRinh172, 10 µM; Sigma Aldrich) were added to activate or inhibit its activity, respectively.

### 2.10. Surface Plasmon Resonance (SPR)

SPR was used to estimate the total amount of the p.Phe508del-CFTR protein in CFBE41o-cells before and after treatment; the expression is too low to be detected using Western blot. We studied the expression of the fully glycosylated form of CFTR (Band C) in untreated and MBTP1-inhibitor-treated cells, according to the previously described method [39]. Real-time detection of the Band C of CFTR in cell lysates was performed using a Biacore system (Biacore 3000, GE Healthcare, Velizy Villacoublay, France) and its Control Software version 3.2. Injections were performed at 25 °C in HBS-P 1× running buffer (GE Healthcare), at a flow of 5 µL/min. Sensorgrams were analyzed using the BIAevaluation software 4.1.1. For each sample, the indicated RU value is the value on the active flow cell (FC) minus the value of the reference FC, 20 s after the beginning of the dissociation phase. In brief, the experiments were performed in a « sandwich » format, as follows: the CFTR antibody (M3A7, Merck) was covalently linked onto a CM5 sensor chip according to Biacore recommendations, in order to reach 6000 Resonance Units (RU). Various quantities (0.5 to 10 µg) of cell lysates were injected over the antibody to determine which quantity was necessary to reach saturation of the antibody. When the anti-CFTR antibody was saturated with the CFTR present in the cell lysates, Wheat Germ Agglutinin (WGA, Triticum vulgaris, Calbiochem, Wilmington, DE, USA) was injected and the RU values were recorded 20 s after the beginning of the dissociation phase. These RU values were used to compare the relative amount of the Band C in the samples. The opposite experiment, in which WGA was linked onto the sensor chip (9000 RU) and the anti-CFTR was used to detect CFTR, was performed. Negative controls were performed using an irrelevant protein. Samples from three different cultures for each condition were analyzed in triplicate.

### 2.11. Gene Expression Analysis Using mRNA Array

Total RNAs were extracted from treated and untreated cells using RNeasy Mini Kits (Qiagen), according to the manufacturer’s instructions. Purity of the RNA was analyzed using the measurement of the 260/280 absorbance with a nanodrop. cDNA was produced from 500 ng of total RNA using an RT² First Strand Kit (Qiagen), according to the manufacturer’s instructions. Gene expression analysis was performed with the RT² Profiler™ PCR Array Human Unfolded Protein Response kit (96-Well Format, PAHS-089Z, Qiagen), using a LightCycler480 (Roche). Supplier protocol was followed for the use of the RT²SYBR Green ROX qPCR Mastermix (Qiagen 330520). Each real-time quantitative PCR (qPCR) plate contained primers specific to 84 genes previously implicated in the UPR, housekeeping genes, and appropriate controls. RT2 Profiler PCR Array Data Analysis was performed according to the manufacturer’s instructions. Differential gene expression was calculated as fold difference using the ΔΔCt method. Fold difference was based on normalization using the average of the three most stable housekeeping genes (HKG; GAPDH, ACTB and B2M). For each of the 84 genes of interest (GOI), the ΔCt value was calculated as ΔCt = CtGOI/CtHKG. The relative fold change in expression was calculated as 2^−ΔCt^. A threshold of four fold changes was chosen for the overexpressed genes. The threshold was two for down-regulated genes. Differentially expressed genes were submitted to a bibliographic analysis, and only the genes with a described or possible link to CFTR are presented here.

### 2.12. Ussing Chamber Recordings

Human tissue from three donors (p.Phe508del/p.Phe508del) was collected and used. Bronchi were incubated in the presence of antibiotics (ceftazidime, 100 µg/mL; vancomycin, 100 µg/mL; tobramycin, 80 µg/mL; Meropenem, 8 µg/mL; amphotericin B 0.25 µg/mL), cleaned, opened, and stored in RPMI-HEPES (plus antibiotic cocktail at 4 °C). They were then incubated twice for 5 min in RPMI-Hepes containing DTT (130 mg/250 mL) to remove mucus, and then rinsed 3 times in RPMI-Hepes. They were then incubated overnight at 4 °C in 9 mL RPMI/HEPES containing 0.05% (*w*/*v*) pronase E and the antibiotic cocktail. After shaking in the presence of pronase E, the bronchi were transferred to a second tube in which 1 mL SVF was added to the supernatant (X4). The bronchi were then discarded and the four supernatants were pooled and centrifuged (5 min, 1200 rpm). The pellet was resuspended in 10 mL PneumaCult, and cells were counted using an automatic counter or a Malassez cell. Cells were then seeded in Petri dishes (10 cm, 1 M cells, P0), previously coated with collagen IV in PneumaCult, in the presence of antibiotics. The medium was changed daily. Amphotericin B and antibiotics were removed after 4–5 days of culture. At confluence, cells were rinsed in PBS/EDTA, trypsinized, counted, and centrifuged (5 min, 1200 rpm). Three plates (control cells, cells treated with the inhibitor for 24 h, and cells treated with the inhibitor for 48 h) of six Snapwells, clear (0.4 µm pore size, 12 mm diameter) and previously coated with collagen IV, were seeded (0.1 M cells) in PneumaCult Ex medium for Ussing chamber measurements. Inserts containing the three pseudo-epithelia were then mounted in a Ussing chamber system (Physiologic Instruments, Reno, NV, USA) composed of two hemi-chambers filled with (in mM) 1.2 NaCl, 115 Na-gluconate, 25 NaHCO_3_, 1.2 MgCl_2_, 4 CaCl_2_, 2.4 KH_2_PO_4_, 1.24 K_2_HPO_4_, 10 mannitol (pH 7.4) for apical solution, and 115 NaCl, 25 NaHCO_3_, 1.2 MgCl_2_, 1.2 CaCl_2_, 2.4 KH_2_PO_4_, 1.24 K_2_HPO_4_, 10 glucose (pH 7.4) for basal solution. Apical and basal solutions were maintained at 37 °C and gassed with 95% oxygen—5% CO_2_. Short-circuit currents (Isc) were measured.

### 2.13. Interleukin 8 (IL-8) Release

Epithelia were reconstituted with cells isolated from homozygous p.Phe508del CF patients (MucilAir™-CF; Epithelix, Plan-les-Ouates, Switzerland). MucilAir™ is a reconstituted human 3D epithelium from airways’ surgical pieces. Cultures were performed at the air–liquid interface, and the mature MucilAir™ was composed of basal cells, ciliated cells, and mucus cells. The release of IL-8 was measured using an ELISA assay (BD Biosciences, Le Pont de Claix, France 555244; detection: 3–200 pg/mL) in the basolateral medium of cells exposed to MBTP1 inhibitor (5, 50, and 100 µM), according to the manufacturer’s instructions. Each ELISA plate contained a standard curve. Absorbance was measured at 450 nm, and the results were normalized to a 24 h secretion. The positive control of inflammation was Cytomix (0.2 mg/mL LPS, 500 ng/mL TNFα, 1% FBS).

### 2.14. Statistical Analysis

Results are expressed as mean ± standard error of the mean (SEM). Differences between experimental groups were evaluated using a two-tailed unpaired Student’s t test and were considered statistically significant when *p* < 0.05 (*), *p* < 0.01 (**), and *p* < 0.001 (***). GraphPad Prism 6.01 software was used.

## 3. Results

### 3.1. Cell Viability and Inflammatory Response

The CFBE41o-cells viability after treatment with the MBTP1 inhibitor was assessed using an MTT test, and, as previously described, cell viability was not significantly altered by the drug (10 µM, 48 h) [33]. Il-8 release was assessed in reconstituted human 3D epithelium from airways’ surgical pieces when 0, 5, 50, and 100 µM PF were applied. As shown in Figure 1, a significant inflammatory response was observed at 100 µM PF which was 10 times more than what was used in the experiments.

### 3.2. MBTP1 Inhibition Inactivates ATF6 and SREBP2 in CFBE41o-Cells

The effect of the inhibition of MBTP1 upon the activation of ATF6 was studied with immunofluorescence, using an anti-ATF6 antibody. The labeling of PDI was used to delimit the ER, and DAPI was used to label the nuclei. Before PF treatment (Figure 2A), ATF6 was observed within the nucleus of the cells (upper image); PDI was observed around the nucleus in the ER (middle image). The merge image (lower image) indicated that ATF6 is mostly present in the nucleus, as well as in the ER of the cells. After treatment (Figure 2B), the localization of ATF6 was modified. As shown in the upper image, ATF6 was no longer visible in the nuclei. Instead, it was co-localized with PDI (middle image), indicating that the treatment maintained ATF6 in the ER, in an uncleaved and inactive form (lower image).

SREBP2 is a protein which is translocated to the nuclei of the cells after its cleavage with MBTP1, and it is used as a marker of the inhibition of MBTP1 and, thus, of the cleavage of ATF6 which is more difficult to detect due to its low expression and poor specificity of the commercially available antibodies. Because the activated form of SREBP2 is in the nuclei, we used nuclei enriched samples. As shown in Figure 3, the cytosolic protein aldolase was present in the cytosolic fraction (lane 1) and in the total protein fraction (lane 6). Its expression was lower in nuclei enriched fractions (lanes 2 to 5). In the presence of the UPR triggering drug thapsigargin, aldolase expression was increased (lane 4) as previously described, indicating the efficiency of the drug [40]. This expression of aldolase was decreased in the presence of the MBTP1 inhibitor (lane 5) suggesting that the increased expression of aldolase when UPR is triggered is under the dependence of ATF6. To further show the nuclei enrichment of our samples, the nucleic protein histone H1 was detected. It was almost absent of the cytosolic and total protein fractions. It was mostly detected in the nuclei enriched samples. The pattern of the detection of aldolase and histone H1 ensured that samples aimed at detecting SREBP2 were indeed enriched in nuclear proteins. Western blot using samples enriched in nuclei (lanes 2 to 5) showed that SREBP2 was absent in the presence of PF (lanes 3 and 5), while it was, even very slightly, observed in the presence of thapsigargin or without any treatment (lanes 2 and 4). Therefore, the inhibition of MBTP1 was obtained even when the UPR was induced in the cells with thapsigargin.

### 3.3. MBTP1 Inhibition Does Not Trigger the UPR in CFBE41o-Cells

The main marker of the triggering of the UPR is the overexpression of its sensor Grp78. Therefore, we assessed its expression after MBTP1 inhibition. Proteins from non-treated, PF-treated, thapsigargin-treated, and PF-plus-thapsigargin-treated cells were loaded on gels. Thapsigargin was used to trigger the UPR, and the PF plus thapsigargin condition was used to assess the putative protection of the MBTP1 inhibitor against the UPR. As shown in Figure 4A, the inhibition of MBTP1 decreased the expression of Grp78 in all conditions, with or without thapsigargin. The lower image shows the detection of actin which was used to normalize the signals for the statistical analysis. The results of the statistical analysis are presented in Figure 4B. When MBTP1 was inhibited, the expression of Grp78 was significantly decreased when compared to non-treated cells and to cells in which UPR was triggered. The expressions of the effectors of the UPR, namely CHOP, IRE1, PERK, and XBP1 were studied using conventional PCR. As shown in Figure 4C, the mRNA of IRE1, PERK, and CHOP were expressed in non-treated cells as well as in treated cells. No significant difference in the expression of the mRNA of CHOP, IRE1, PERK, and XBP1 was observed in the absence or in the presence the MBTP1 inhibitor and was confirmed using statistical analysis (Figure 4D).

### 3.4. MBTP1 Inhibition Increases the Cl-Efflux in Cells Expressing p.Phe508del-CFTR

In order to assess the effect of the inhibition of MBTP1 on the Cl-channel function of p.Phe508del-CFTR, patch-clamp experiments were performed. Currents were measured in non-treated cells (Figure 5A) in the presence of the CFTR’s activators (Forskolin and Geneistein) and in the presence of the inhibitor-172, in order to verify that the recorded currents were due to p.Phe508del-CFTR. As expected, currents increased in the presence of the activators and decreased when the inhibitor was added. We next compared the effect of the inhibitor of MBTP1 with the control condition with currents due to the presence of VX-809 that was used as a reference and with a combination of PF plusVX-809. Representative I/V curves are shown in Figure 5B. A statistical analysis was performed after a normalization of the currents (pA/pF) using the current values obtained at +80 mV. A bar graph representation of the results is shown in Figure 5C. A significant increase of the Cl-efflux via the p.Phe508del-CFTR channel was observed after the inhibition of MBTP1. The currents were greater in the presence of PF than in the presence of VX-809. However, no synergistic effect between both molecules was observed.

To verify the effect of the inhibition of MBTP1 in a more relevant model, shot-circuit currents were measured in epithelia from patients using the Ussing chamber method. The statistical analysis showed that the inhibition of MBTP1 increased the currents through p.Phe508del-CFTR, after 24 h and 48 h of treatment (Figure 5D).

### 3.5. Inhibition of MBTP1 Increases the Transcription and Protein Expression of p.Phe508del-CFTR

In order to compare the transcription of p.Phe508del-CFTR in cells before and after the inhibition of MBTP1, real-time quantitative PCRs were performed. Because we failed to obtain reliable results with native cells, we used CFBE41o-/F508del cells. Figure 6A is the bar graph representation of the statistical analysis of the 2^−ΔCt^ using β-Actin as a housekeeping gene and non-treated cells as the reference. The results show that the inhibition of MBTP1 significantly increased the gene expression of p.Phe508del-CFTR.

We further studied the protein expression of p.Phe508-del-CFTR in both CFBE41o-(Figure 6B) and CFBE41o-/F508del cells (Figure 6C) using immunoprecipitation. In both cell lines, we observed an increased Band-B expression after inhibition of MBTP1. Nevertheless, after VX-809 treatment, the amounts of Band B and Band C were higher than with the inhibitor. The combination of PF and VX-809 likely had an additive effect on the synthesis of Band B.

### 3.6. Inhibition of MBTP1 Increases the Expression of p.Phe508del-CFTR Protein

Because the expression of the p.Phe508del-CFTR protein is very low in non-transduced CFBE41o-cells, we used SPR to assess its amount before and after the inhibition of MBTP1. Cell lysates (2.5, 5, and 10 µg) were injected over the immobilized antibody directed against CFTR. The association, equilibrium, and dissociation phase were observed and the RU values used for the statistical analysis were recorded. The bar graph representation of the analysis is shown in Figure 7A. For each quantity of the total protein that was injected, the amount p.Phe508del-CFTR protein which remained linked onto the anti-CFTR antibody was significantly higher in lysates of treated cells, indicating that inhibiting the activity of MBTP1 increases the total amount of the p.Phe508del-CFTR protein in the cells.

In order to know which form of CFTR was overexpressed (Band B or C), we used a previously described method in which WGA is used to specifically link the C band of CFTR [39]. WGA was immobilized and the proteins were injected. An anti-CFTR was then injected to ensure that the linked proteins onto the WGA were CFTRs. CFBE41o-/F508 cells were used to obtain enough p.Phe508del-CFTR protein in the lysate for these experiments. CFBE41o-/corr cells were used as a positive control for the expression of the mature form of CFTR, and the linearity of the response was assessed The results of the SPR analysis are presented in Figure 7B. The upper curves represent the RU values of the sandwich formed by WGA/CFTR/anti-CFTR for various quantities of injected proteins. A plateau phase is observed above 2.5 µg of the injected proteins. A statistical analysis of the responses was performed and is presented in Figure 7B, in the lower panel. The amount of the mature form of CFTR was significantly higher in cells expressing the normal CFTR than in cells expressing the p.Phe508del-CFTR protein, with or without MBTP1 inhibitor. The comparison between cells expressing the p.Phe508del-CFTR protein before and after PF treatment did not show any difference. In the reverse experiment, in which the anti-CFTR antibody was linked, the proteins were injected, and the WGA was performed. As shown in Figure 7C, in the upper panel, the presence of the mature form seemed increased in the lysates of the cells expressing the p.Phe508del-CFTR protein after MBTP1 inhibition. Nevertheless, the statistical analysis (Figure 7C, lower graphs) did not confirm this observation.

The conclusion of the SPR experiments is that MBTP1 inhibition increases the total amount of p.Phe508del-CFTR protein and that the increased form of CFTR is in Band B.

### 3.7. PF-429242 Increases the Expression of p.Phe508del-CFTR in Membranes

After showing that the inhibition of MBTP1 increases the global RNA and protein levels of p.Phe-508del-CFTR, we focused on its expression within membranes. Figure 8 shows a representative image of a blot performed with crude membrane protein enriched samples (lanes 1 to 4) and with total lysate (lane 5). We observed that PF strongly increased the expression of Band B in the membranes, whereas VX-809 increased the expression of Band C. The combination of PF and VX-809 induced increased expression of both Band B and C in the membranes. The middle and the lower panels in Figure 8 show the detection of the Na+/K+-ATPase and aldolase in the samples, indicating that our samples were indeed enriched with membrane proteins.

### 3.8. Comparison of the Gene Expression between Non-Treated and PF-429242-Treated Cells

The Human Unfolded Protein Response RT² Profiler PCR Array used in the present study profiles the expression of 84 key genes involved in the UPR. It also determines whether the UPR pathway activity is increased or unchanged in experimental samples. We arbitrarily selected a selective modulation threshold of four in order to isolate genes that exhibit huge changes in their expression in comparison to controls. A scatterplot showing the gene distribution of each gene modulation by comparing data from untreated and treated cells is shown in Figure 9. Most of the genes were up-regulated with the treatment of the cells with PF. According to our selective criteria, seven genes were found to be overexpressed, and three genes were downregulated when MBTP1 was inhibited. The significantly overexpressed genes were HSPA1B (Heat Shock Protein Family A (Hsp70 member 1B; fold change: +4.87; *p* < 0.05), CEBPB (CCAAT Enhancer Binding Protein Beta; fold change: +4.51; *p* < 0.05), VIMP (VCP-Interacting Membrane Protein; fold change: +4.18; *p* < 0.001). The PFND2, MAPK8, XBP1 and PPP1R15A genes had fold changes of 4.1, 4.18, 4.37, 4.51, 4.87, 5.08, and 7.76, respectively. Nevertheless, their increased expression was not found to be significant. The downregulated genes (threshold 2) were INSIG1 (Insulin Induced Gene 1; fold change: −1.94; *p* < 0.005), CALR (Calreticulin; fold change: −4.9; not significant) and HSPA5 (BiP; fold change: −2.1; not significant).

## 4. Discussion

The triggering of the UPR in CF is still in debate. Some authors indicate that it is triggered by inflammation, and others indicate that it is likely triggered by the misfolded CFTR [29,31] leading to an atypical form of UPR [32]. UPR is a complex process involving misfolded proteins, inflammation, and also infection, which are all present in CF [41]. Whatever triggers the UPR in CF, it inhibits endogenous CFTR expression due to the activation of ATF6 [29,30,42]. To be activated, ATF6 has to be cleaved with the serine protease MBTP1 [24]. Therefore, we hypothesized and showed that the inhibition of this enzyme restores Cl-efflux in cells overexpressing p.Phe508del-CFTR [33]. The aim of the present study was to show that in more relevant cells for CF (cells endogenously expressing p.Phe508del-CFTR and human bronchial cells from patients), the inhibition of MBTP1 alleviates the defects due to p.Phe508del-CFTR and highlights the involved mechanisms.

We found that the inhibition of MBTP1 did not induce cell mortality or inflammation. The inhibition impedes the activation of ATF6 and SREBP2, and, subsequently, it increases the expression of p.Phe508del-CFTR within membranes, in accordance with our previous results obtained using different cell models than the ones used here [29,33]. Therefore, we showed for the first time that the inhibition of MBTP1 alleviates the defects due to p.Phe508del-CFTR in relevant models for CF.

Inhibition of MBTP1 presents a promising avenue for therapeutic intervention across diverse medical conditions. In cancer, targeting MBTP1 impedes the processing of SREBPs, thereby disrupting lipid metabolism crucial for cancer cell proliferation [43,44]. In dyslipidemia, inhibiting MBTP1 may offer a means to modulate lipid homeostasis [45,46]. Additionally, the emerging research suggests that MBTP1 inhibition could be explored as a strategy against viral infections, as this protease plays a role in viral replication processes [47,48]. The multifaceted potential of MBTP1protease inhibition underscores its significance as a therapeutic target. While MBTP1 inhibition is proposed as a potential therapy for various diseases, its pleiotropic nature raises concerns about its therapeutic use. Indeed, it is involved in diverse cellular processes, including lipid metabolism, immune response, and cellular stress. The broad impact of its modulation may result in unintended consequences, affecting physiological functions beyond the targeted disease pathway. Striking a balance between therapeutic efficacy and minimizing off-target effects poses a challenge. A comprehensive understanding of the intricate roles MBTP1 plays in cellular pathways is crucial for optimizing its therapeutic potential while mitigating adverse effects. Indeed, the MBTP1 inhibition may have off-target effects that could be depicted with the inactivation of the MBTP1 gene. Nevertheless, for male CD1 mice treated with the inhibitor of MBTP1 at doses of 10 and 30 mg/kg/dose i.p. every 6 h for 24 h, beside the altered lipogenesis due to the inhibition of SREBP, no side effects were reported [36,49]. Interestingly, rare cases of patients with reduced MBTP1 expression due to bi-allelic pathogenic variants resulting in MBTPS1 deficiency have been reported [50,51]. These patients presented skeletal dysplasia and increased circulating lysosomal enzymes, which is by far the most obvious clinical manifestation, but exhibited no pulmonary alterations. These results suggest that, if MBTP1 inhibition were to be used in CF patients, precautions should be taken regarding dosage and duration of treatment.

In order to further estimate the effect of the inhibition of MBTP1 upon the triggering of the UPR, we assessed the expression of the UPR’s sensor Grp78, which is overexpressed when the ER is overloaded. Without, or in the presence of thapsigargin, which is a UPR inducer, we observed a decreased expression of the UPR’s sensor when PF was applied, indicating that UPR is decreased when MBTP1 is inhibited. Another possible explanation for the decreased expression of Grp78 is that the inhibition of ATF6 leads to a decreased presence of the p.Phe508del-CFTR protein in the ER that offers no binding sites for Grp78, inhibiting the UPR triggering. This could be due to an increased ERAD activity or to an increased output of the protein from the ER. Because the UPR induces the overexpression of the effectors of the ERAD with its three arms, we assessed the expression of IRE1 and XBP1, PERK, and CHOP. Under PF treatment, no modification of the expression of these effectors was observed. Because when IRE1 is activated it splices the mRNA encoding XBP1 which encodes a bZIP transcription factor that activates the expression of enzymes for the degradation of misfolded proteins [52,53], the absence of its activation likely indicates that the p.Phe508del-CFTR protein is less degraded and thus likely exported out of the ER. It has to be noted that the activation of XBP1 has, partially, a redundant role with ATF6 [54]. Since the PERK pathway leads to the attenuation of the translation of proteins and to the ER stress-induced apoptosis [22], the absence of its increased expression suggests that the stress-induced protein translation attenuation, as well as the apoptosis programs due to CHOP, is not prevalent when MBTP1 is inhibited. Nevertheless, XBP1s were detected, without any increased expression of IRE1. These results are in accordance with previous works, showing that ATF6 leads to an upregulation of XBP1 [55]. Nevertheless, the amount of XBP1 mRNA is a rate-limiting factor in the production of XBP1S [55]. Because the mRNA of XBP1 is not increased in the presence of PF, the spliced XBP1 produced at a low level should be rapidly degraded by the proteasome, as previously described [53]. Taken together, these results indicate that the inhibition of MBTP1 does not trigger the UPR and that it is likely due to the exit of p.Phe508del-CFTR from the ER to the plasma membrane.

Our assumption that the inhibition of MBTP1 allows misfolded proteins to exit the RE and reach the membrane was further tested. We observed increased Cl-efflux via the mutated CFTR, showing that there is more p.Phe508del-CFTR protein in the membrane of the cells. Furthermore, we showed that the p.Phe508del-CFTR’s mRNA and protein are increased. Finally, we assessed the presence of the p.Phe508del-CFTR protein within membranes and found that, when MBTP1 is inhibited, there is a higher synthesis of p.Phe508del-CFTR protein and that the alleviation of the Cl-efflux is due to an increased amount of the immature form of CFTR (Band B) in membranes, which is also active as a Cl-channel [11]. This result also indicates that the observed rescue does not occur through the Golgi apparatus but probably involves unconventional protein secretion (UPS), which is in accordance with previous results showing that the rescued p.Phe508del-CFTR protein can use UPS [56]. This is a key result because, to our knowledge, no corrector of the p.Phe508del-CFTR protein is able to also induce an increased synthesis of its mRNA and protein. Beside UPS, ATF6 inhibition downregulates the expression of genes involved in the ER-associated degradation (ERAD) pathway. These genes are Derlin-1, which is involved in the retrotranslocation of misfolded proteins from the ER to the cytosol for subsequent degradation; Homocysteine-induced ER protein, which is involved in the degradation of misfolded proteins; ER degradation-enhancing alpha-mannosidase-like protein, which is involved in the recognizing and targeting misfolded glycoproteins for degradation; p97/VCP, which plays a crucial role in extracting ubiquitinated substrates from the ER membrane for subsequent degradation with the proteasome; and SEL1L, which forms a complex with the ERAD E3 ubiquitin ligase and contributes to the recognition and degradation of misfolded proteins. Therefore, if ERAD activation is decreased or impaired, there might be a reduced efficiency in targeting F508del-CFTR for degradation. As a result, a larger fraction of the immature (Band B) F508del-CFTR may escape degradation and progress through the secretory pathway. Aside from degradation, it is important to note that the exact impact of ATF6 on endocytosis and recycling membrane proteins is not fully understood and that changes in its activation may modulate the internalization or recycling of specific membrane proteins, which are increased in the case of F508del-CFTR.

To further understand how the inhibition of MBTP1 can rescue the p.Phe508del-CFTR protein and because the active form of ATF6 is a transcription factor targeting many genes, we used a qPCR-Array of 84 genes. The significantly overexpressed genes were HSPA1B, CEBPB, and VIMP. HSPA1B (Heat Shock 70 KDa Protein 1B) is a molecular chaperone implicated in the protection of the proteome from stress, the folding and the transport of newly synthesized polypeptides, and in the activation of the proteolysis of misfolded proteins. It is involved in the degradation of the p.Phe508del-CFTR protein [57]. Nevertheless, HSPA1B maintains protein homeostasis during cellular stress using two opposing mechanisms, which are protein refolding and degradation. It is demonstrated that it facilitates protein refolding after stress and slowly evolves to protein degradation depending on its acetylation state [58]. Therefore, in the present work, it is possible that it acts positively to protect p.Phe508del-CFTR. CEBPB is a transcription factor regulating the expression of genes involved in immune and inflammatory responses. It is essential in the lung for the spatial and temporal regulation of the expression of the *CFTR* gene. Because it is demonstrated that CEBPB is a positive regulator acting on the *CFTR* promoter, it is a good candidate to explain our observed enhanced expression of the p.Phe508del-CFTR protein after the inhibition of MBTP1 [59,60]. VIMP is a small protein located at the ER membrane that interacts with both Derlin-1 and VCP, implying that it participates in ERAD. Unfortunately, it cannot explain the rescue of the p.Phe508del-CFTR protein because its overexpression reduces the steady-state level of p.Phe508del-CFTR protein by shortening its half-life [61]. Despite having no significant overexpression, the PFND2, MAPK8, XBP1, and PPP1R15A genes were increased when MBTP1 was inhibited. PFND2 (Prefoldin Subunit 2) is a chaperone involved in protein folding that was shown to favor the rescue of the p.Phe508del-CFTR protein in HEK293 and BHK cells [62]. MAPK8/JNK1 is a serine/threonine-protein kinase involved in various processes such as cell proliferation, differentiation, migration, transformation, programmed cell death, and autophagy. It was shown that the control of the stability of the CFTR mRNA is linked to the phosphorylation state MAPK [62]. Importantly, MAPK8/JNK1 is a downstream effector of the kinase activity of the IRE1α pathway [18]. Some studies describe that the kinase pathway TRAF2-ASK1-JNK1, following the IRE1α activation, leads to the autophagy which seems to be involved in the transport of p.Phe508del-CFTR to the plasma membrane through the UPS pathway [63,64,65]. The role of XBP1 was discussed above. PPP1R15A (GADD34) dephosphorylates the translation initiation factor eIF-2A/EIF2S1 leading to reverse of the shutoff of protein synthesis initiated by stress and facilitating the recovery of cells [66,67,68]. Therefore, its activation is in favor of an increased expression of CFTR in our model. Polyubiquitinated GADD34 is rapidly eliminated upon removal of cell stress, but when the proteasome is silenced, it is stabilized. It was shown that the stabilized form of GADD34 enhanced the accumulation of the p.Phe508del-CFTR protein [68]. Therefore, its overexpression in our treated cells is in favor of an increased expression of the mutated CFTR.

In contrast, INSIG1, CALR, and HSPA5 were found to be decreased, but significance was reached only for INSIG1. INSIG1 is a membrane protein of the ER that regulates cholesterol metabolism and lipogenesis homeostasis through blocking the processing of SREBPs [69]. Nevertheless, the SREBPs are activated with MBTP1, and we found that SREBP2 was inhibited by PF [69,70]. Thus, the decreased INSIG1 expression is further evidence that the inhibition of MBTP1 induces the inhibition of ATF6 in our model. Lipids are unbalanced in CF with an increased accumulation of cholesterol in CF cells, whereas SREBPs are the main regulators of the lipid homeostasis [71,72]. Therefore, their decreased accumulation due to the decreased SREBP2 activity found that when MBTP1 is inhibited this could favor a better membrane insertion and function of p.Phe508del-CFTR [73]. Because it was shown that ER stress responses can promote lipid accumulation in hepatocytes, and despite, to our knowledge, that this was never searched in epithelial cells, it can be proposed that the inhibition of MBTP1 without UPR triggering could be beneficial in CF cells. Interestingly, CALR was decreased even though significance was not reached when MBTP1 was inhibited. This could also explain our results by helping the membrane exportation of p.Phe508del-CFTR because CALR is a negative regulator of the cell surface expression of the mutated CFTR [74].

## 5. Conclusions

In conclusion, we show here for the first time that the inhibition of the MBTP1 enzyme alleviates the p.Phe508del-CFTR defects in cells endogenously expressing the mutated CFTR (cell line and patient’s bronchial cells), with a better effect than that of VX-809. Nevertheless, a triple combination (elexacaftor/tezacaftor/ivacaftor, Trikafta™, Vertex, Boston, MA, USA) is approved for the treatment of CF patients with at least one p.Phe508del mutation or at least one other mutation in the CF gene that is responsive to Trikafta (i.e., 177 other approved mutations), regardless of their second mutation type [75,76,77] (www.cff.org, accessed on 20 November 2023). Whereas Trikafta™ is generally well tolerated with mild adverse events, 10-30% of the CF patients remain without any therapeutic alternative [75,78,79]. Indeed, patients with rare variants remain excluded from the treatment, and no clinical trials supported the use of Trikafta™ in the majority of rare genotypes lacking the p.Phe508del allele [80]. For these rare mutations and for the patients receiving Trikafta but presenting severe side effects, new molecules are needed. Inhibitors of MBTP1, alone or in combination with existing molecules, could be an alternative. Indeed, an increased synthesis and membrane localization of p.Phe508del-CFTR could be beneficial when a functional CFTR protein with decreased ion transfer is expressed, when a decreased production of the protein is observed, and when there is a reduced CFTR membrane stability, which increases the turnover of the protein. Nevertheless, whereas inhibition of MBTP1 seems to be well tolerated in mice, in CF patients, the potential side effects would need to be carefully considered. One approach could involve developing MBTP1 inhibitors that are specifically designed for short-term use during exacerbations. This would aim to harness the potential therapeutic benefits while minimizing the risk of long-term side effects. The therapy could be administered in conjunction with existing treatments during exacerbations to address the acute symptoms. Close monitoring of patients and thorough assessment of both short-term benefits and potential side effects would be essential in evaluating the effectiveness and safety of this approach. As with any potential therapy, further preclinical and clinical research would be necessary to determine the optimal dosage, duration, and safety profile of MBTP1 inhibition in the context of CF exacerbations. Collaborative efforts between researchers, clinicians, and pharmaceutical companies would be crucial to advance the development of such targeted therapies.

## Figures and Tables

**Figure 1 cells-13-00185-f001:**
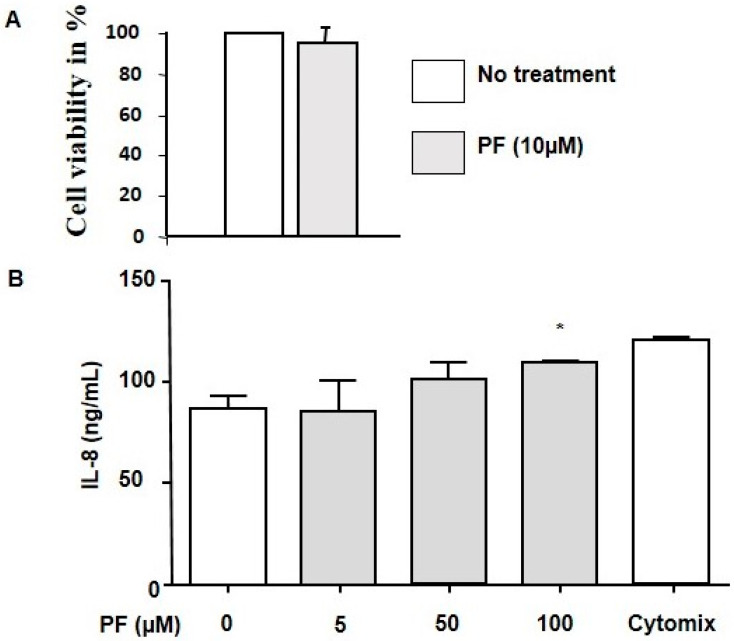
Assessment of the cell viability and the inflammatory response due to MBTP1 inhibition in native CFBE41o-cells and in epithelia from CF patients, respectively. (**A**) Viability was assessed in CFBE41o-cells using an MTT test. No significant decreased viability was observed in the presence of 10 µM PF for 24 h. B. IL-8 secretion was evaluated in epithelia from CF patients after different concentration of PF (0.5 µM, 50 µM, 100 µM; *n* = 3). Cytomix was used as a positive control of inflammation. The bar graph represents the statistical analysis after ELISA assay and indicates that the inhibition of MBTP1 has a significant effect on IL-8 secretion only at 100 µM treatment conditions. (**B**) Bar graphs representing the statistical analysis of the IL-8 release in the presence of PF (0–100 µM). Cytomix was used as a positive control. Significance (*p* < 0.05) was observed in the presence of 100 µM of PF. *: *p* < 0.05.

**Figure 2 cells-13-00185-f002:**
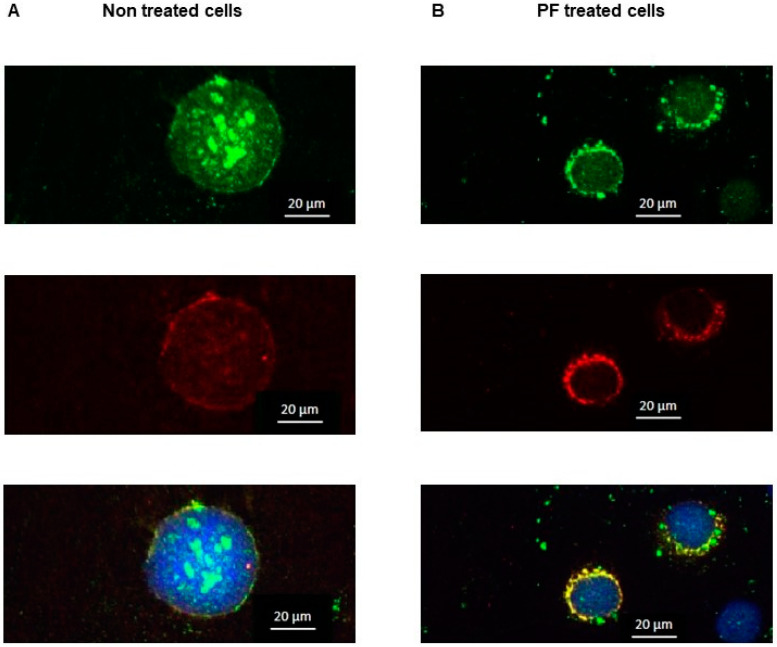
Localization of ATF6 in response to MBTP1 inhibition in native CFBE41o-cells. Immunofluorescence was used to assess the localization of ATF6 before and after treatment with PF. (**A**) In the absence of treatment, ATF6 is in the nuclei of the cells (upper image). The middle image is the labeling of the ER with PDI. In the lower image, which is a merged image of ATF6 and PDI, it can be observed that ATF6 is mainly located in the nuclei which is a hallmark of the UPR triggering. (**B**) When MBTP1 is inhibited, ATF6 is observed around the nuclei (upper image). The middle image is the labeling of the ER with PDI. In the lower image, which is a merged image of ATF6 and PDI labeling, it can be observed that ATF6 is mainly located around the nuclei and is co-distributed with PDI. Therefore, the inhibition of MBTP1 retains ATF6 in the ER which is a hallmark of its non-activation. Each labelling was performed eight times, and the images are representative of 150–200 cells by field.

**Figure 3 cells-13-00185-f003:**
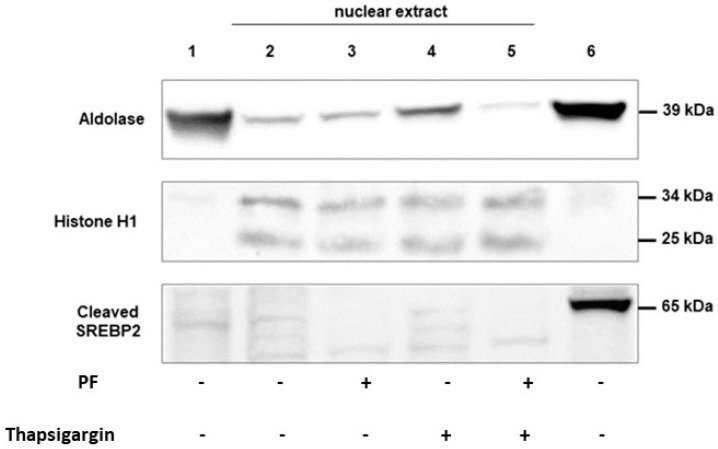
Localization of SREBP2 in response to MBTP1 inhibition in native CFBE41o-cells. The localization of SREBP2 was assessed using Western blot in cytosolic extract (lane 1), nuclei enriched samples (lanes 2 to 4), and total lysate lane (6). To ensure the enrichment of nuclear proteins, aldolase was used as a cytosolic marker (**upper panel**) and Histone H1 as a nuclear marker (**middle panel**). SREBP2’s cleaved form (**lower panel**) was absent in nuclear extract of treated cells indicating that its cleavage was abolished.

**Figure 4 cells-13-00185-f004:**
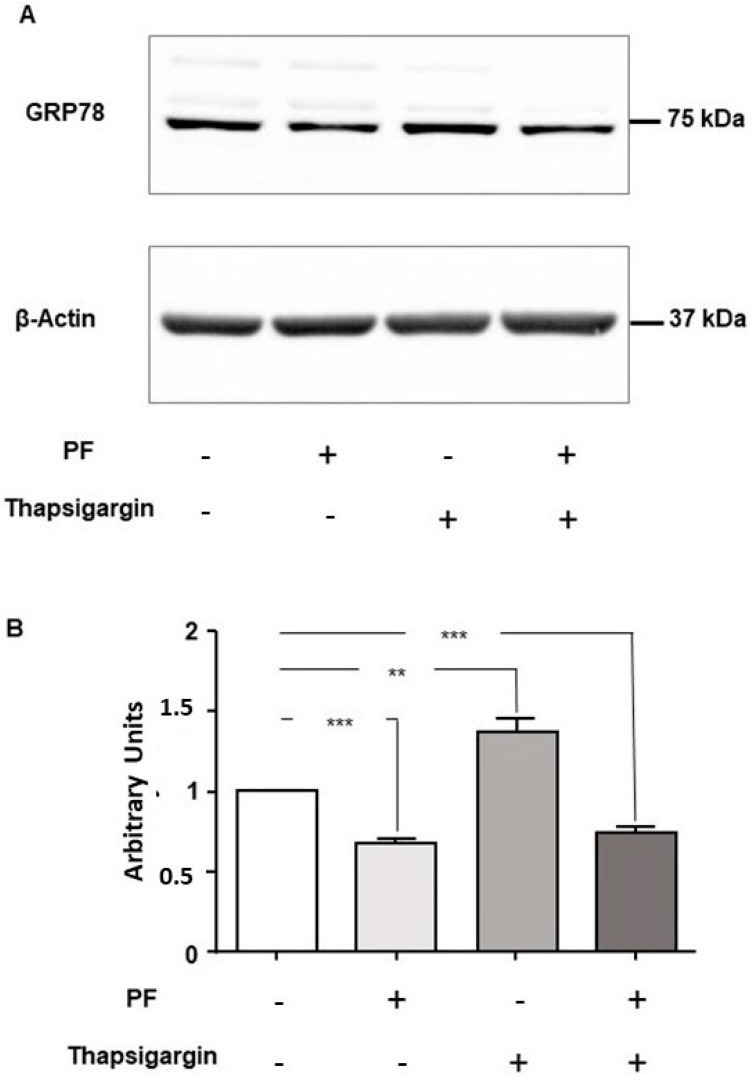
Expression of the UPR markers in response to MBTP1 inhibition in native CFBE41o-cells. (**A**) The expression of Grp78 with or without PF and with or without thapsigargin was analyzed using Western blot (*n* = 4). The image is a representative image of the detection of Grp78 (**upper panel**) and of β-actin (**lower panel**). A higher expression of Grp78 is observed in the presence of thapsigargin when compared to non-treated cells. A lower expression of Grp78 is observed when MBTP1 is inhibited, with or without thapsigargin. (**B**) Bar graph representation of the statistical analysis of the expression of Grp78 which is significantly increased in the presence of thapsigargin and significantly decreased when MBTP1 is inhibited, with or without thapsigargin. (**C**) Representative gels worked to detect and quantify the mRNA of CHOP, IRE1, PERK, and XBP1 using PCR. NT: non-treated cells, PF: PF treated cells, NTC: no template control. (**D**) The bar graphs are statistical representations of the mRNA level of CHOP, IRE1, PERK, and XBP1. No significant modification of their expression was observed., *p* < 0.01 (**), and *p* < 0.001 (***).

**Figure 5 cells-13-00185-f005:**
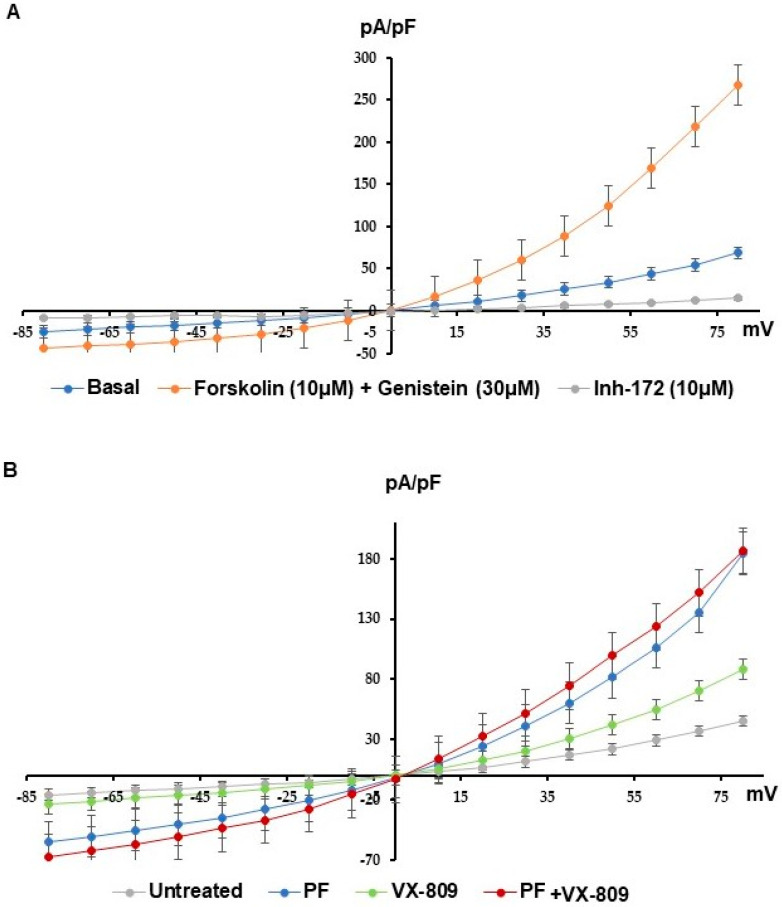
Cl-efflux when MBTP1 is inhibited in native CFBE41o-cells. Cl-efflux via the p.Phe508del-CFTR channel in the presence of PF with or without VX-809 were assessed using patch clamp (whole-cell configuration). (**A**) Representative I/V curves were obtained in a basal state with CFTR’s activators and inhibitors which were used to assess the specificity of the recorded currents. (**B**) Representative I/V curves were obtained with PF and/or VX-809. Increased currents via p.Phe508del-CFTR were observed in all conditions when compared to the control condition. The conditions with PF show higher currents than other conditions. (**C**) The bar graph represents the statistical analysis (Untreated: *n* = 5; PF: *n* = 10; VX-809: *n* = 5; PF plus VX-809: *n* = 4) of the normalized currents recorded at +80mV. MBTP1 inhibition significantly increases p.Phe508del-CFTR channel currents above more than that of VX-809, but no significant synergistic effect is observed. (**D**) Example of curves recorded during the Ussing chamber experiments. The upper curve was obtained with PF-treated bronchial epithelial cells from a patient. The lower curve is the recording made with non-treated cells from the same patient. The responses to CFTR’s activators and inhibitor were enhanced with PF. (**E**) The bar graph represent the statistical analysis of p.Phe508del-CFTR currents recorded on bronchial epithelia from a homozygous p.Phe508del patient in Ussing chamber assays. We show that MBTP1 inhibition (24 and 48 h) significantly increases the p.Phe508del-CFTR currents in comparison to the control condition (*n* = 3). *p* < 0.01 (**), and *p* < 0.001 (***).

**Figure 6 cells-13-00185-f006:**
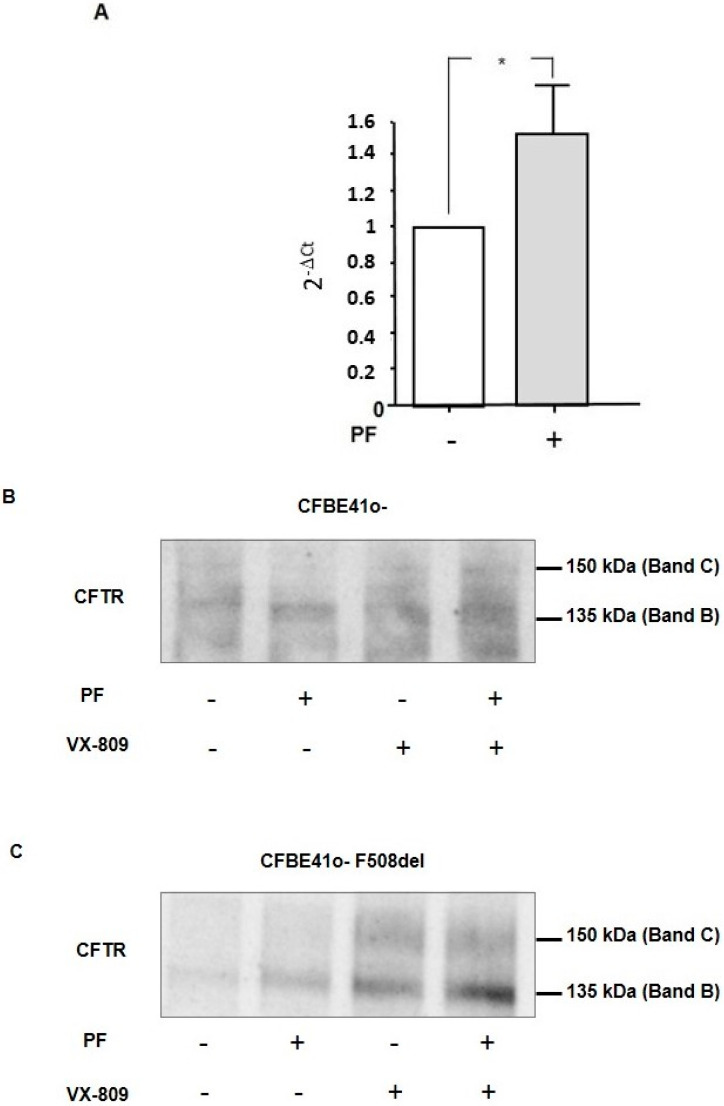
Expression of the CFTR mRNA and protein when MBTP1 is inhibited in native CFBE41o-cells and in transduced cells. (**A**). The bar graph is the representation of the quantification of the real-time quantitative PCRs performed in CFBE41o-/F508del cells, showing that the inhibition of MBTP1 significantly increases the gene expression of p.Phe508del-CFTR. Representative Western blot analyses after CFTR immunoprecipitation in CFBE41o-cells and in transduced CFBE41o-/F508del cells are shown in (**B**,**C**), respectively. VX-809 was used to compare the qualitative effect of PF for the global p.Phe508del-CFTR synthesis. In CFBE41o-and CFBE41o-/F508del, the inhibition of MBTP1 increases the expression of Band B of CFTR. VX-809 also increase the CFTR Band-B expression, and we found a synergistic effect on Band B expression using a combination of PF and VX-809. *p* < 0.05 (*).

**Figure 7 cells-13-00185-f007:**
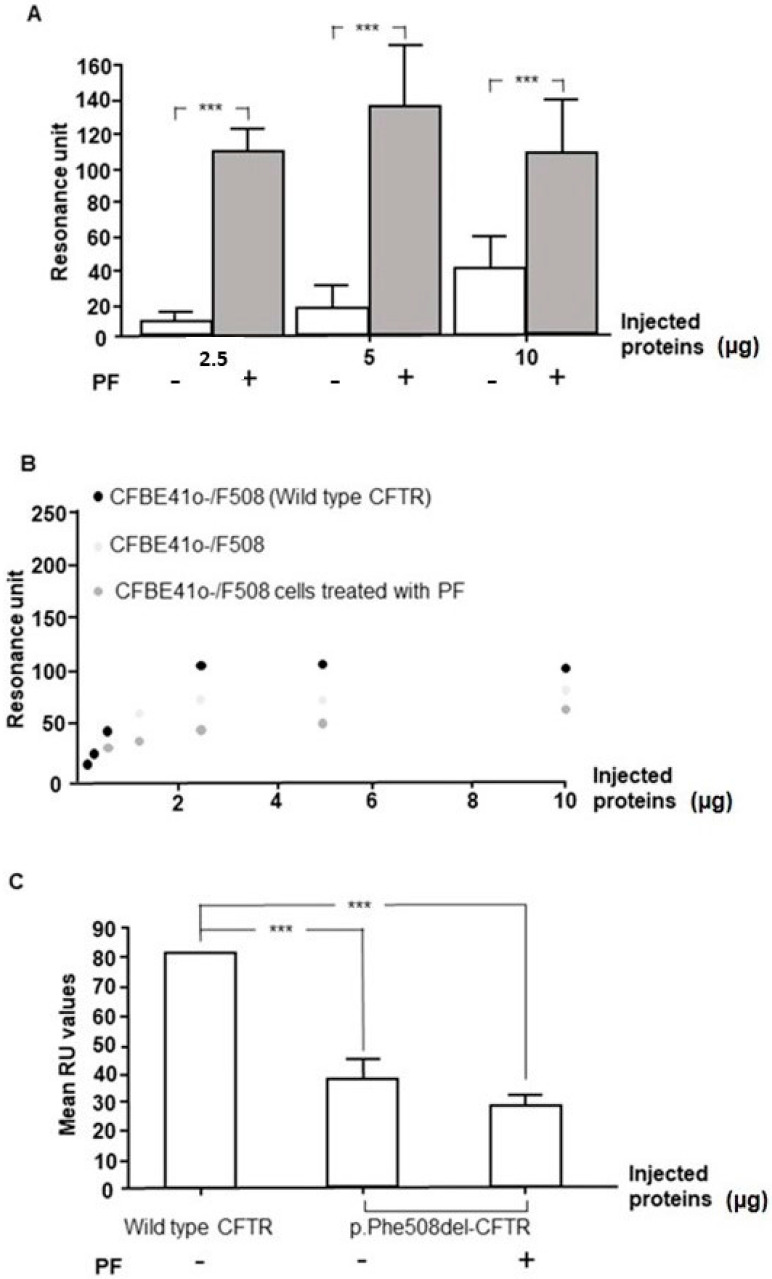
Study of the expression of the p.Phe508del-CFTR protein using SPR in native CFBE41o-cells. The expression of p.Phe508del-CFTR was assessed without and in the presence of PF. (**A**) Amounts of 2.5, 5, and 10 µg of total proteins from cell lysates were injected over an immobilized anti-CFTR antibody. The RU values were recorded 20 s after the beginning of the dissociation phase and used for the statistical analysis (*n* = 3). The bar graph represents the analysis and shows that the total amount of the p.Phe508del-CFTR protein is significantly increased when MBTP1 is inhibited. (**B**) To assess which form of p.Phe508del-CFTR used, we used WGA to specifically link the Band C and an anti-CFTR antibody to ensure that the linked proteins onto the WGA are really CFTRs. The upper panel represents the curves of the RU values in function of the total amount of injected proteins (from 0 to 10 µg). The lower bar graph (**C**) represents the analysis of the values obtained with 10 µg of proteins (*n* = 5) and shows that there is less CFTR protein in cells expressing the mutated CFTR than in cells expressing the Wt-CFTR. In cells expressing p.Phe508del-CFTR, no more protein was linked onto WGA in the presence of PF. (**D**) The opposite analysis in which the anti-CFTR was linked and WGA was injected was performed (*n* = 5). The statistical analysis (**E**) show that the results are identical to those observed in (**B**). *p* < 0.001 (***).

**Figure 8 cells-13-00185-f008:**
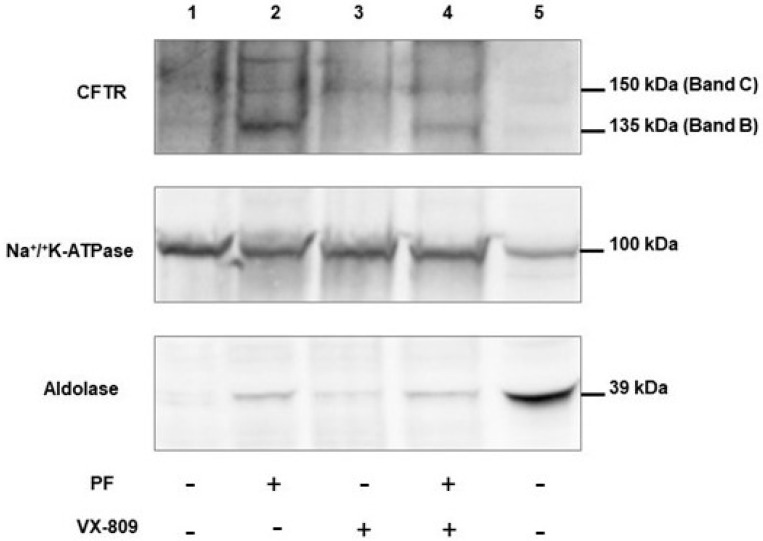
Membrane expression of p.Phe508del-CFTR when MBTP1 is inhibited in native CFBE41o-cells. The upper image is a representative Western blot showing the expression of p.Phe508del-CFTR in membrane extract (lanes 1 to 4) and in total lysate (lane 5) when cells are treated with PF and/or VX-809. Na^+^/K^+^-ATPase was used as a marker of membranes (middle panel), and aldolase was used as a cytosolic marker (lower panel). The inhibition of MBTP1 increases the membrane expression of Band B of CFTR, and the association with VX-809 increases Band C of CFTR at the membrane.

**Figure 9 cells-13-00185-f009:**
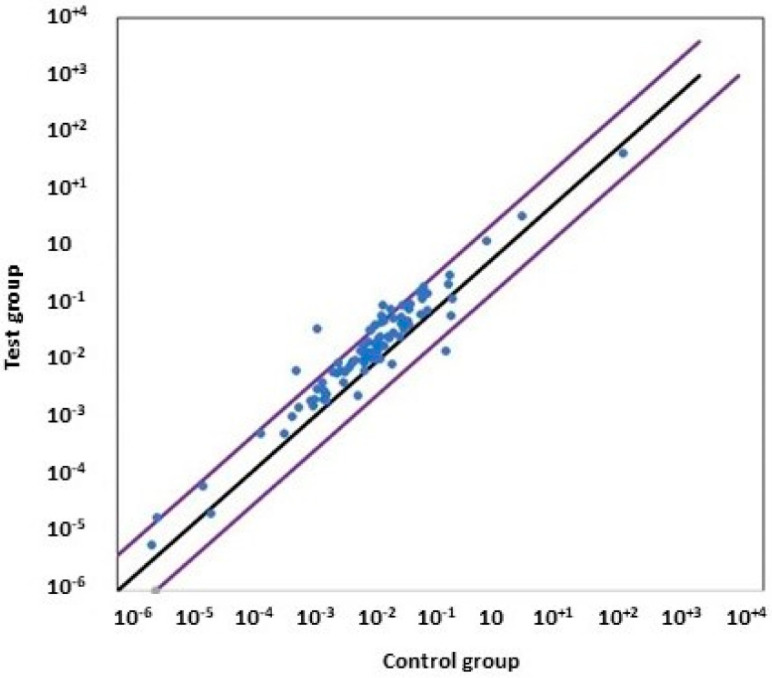
Gene modulation when MBTP1 is inhibited in native CFBE41o-cells. Untreated and treated cells were subjected to a qPCR-Array (*n* = 6) aimed to assess 84 genes involved in the UPR (blue dots). The image is a scatterplot showing the gene distribution of each gene modulation by comparing data from untreated and treated cells. The x axis is the 2^−Δct^ of the control group, and the y axis is the 2^−Δct^ of the test group. Most of the genes were upregulated when MBTP1 was inhibited. According to our selective criteria delimited with purple diagonal bars, seven genes were found to be overexpressed, and three genes were found to be downregulated. The black diagonal represents the absence of modulation.

## Data Availability

Data are contained within the article.

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
