# Peer review of "The Inhibition of the Membrane-Bound Transcription Factor Site-1 Protease (MBTP1) Alleviates the p.Phe508del-Cystic Fibrosis Transmembrane Conductance Regulator (CFTR) Defects in Cystic Fibrosis Cells"

_cells, 2024, doi:10.3390/cells13020185_

Round 1
Reviewer 1 Report
Comments and Suggestions for Authors
See above
Reviewer 2 Report
Comments and Suggestions for Authors
The manuscript by Santinelli et al builds on previous work that demonstrated ATF6 inhibition increased F508del CFTR processing and activation. This manuscript extends this mechanism another step by demonstrating that inhibition of the protease MBTP1, an activator of ATF6, has similar outcomes to ATF6 inhibition. Specifically, band B expression of F508del CFTR is elevated and CFTR function is increased. The study is thorough and analyzes several aspects of CFTR processing with the inhibitor with and without VX809, as well as broader impacts of MBTP1 inhibition on gene expression. Mechanistically this work only provided an incremental extension, but the results are consistent with previous work and the incremental extension is important.
Major comments
1. Quality of some western blots is low, though admittedly, CFTR can be difficult to probe. Isolating membranes in figure 8 is a benefit to the study to help enrich membrane bound CFTR.
2. More discussion as to why only band B is enriched effectively by PF is needed. Though there are mechanisms by which band B can get to the membrane and function, those are secondary pathways of CFTR trafficking and generally affects only small amounts of CFTR. The authors’ views on how the preference of band B correlates to the functional gains would be a benefit. The authors mention the UPS, but more discussion on whether this is sufficient for a clinical benefit would be interesting.
3. In figure 5, only pooled data are presented. An example of the Ussing traces showing gains in function are needed.
4. More discussion on the therapeutic potential for MBTP1 inhibition is needed. What are the risks associated with broadly inhibiting ATF6 function? What are the other targets of MBTP1?
The manuscript is straightforward and extends a previously published mechanism of CFTR regulation.
Round 2
Reviewer 1 Report
Comments and Suggestions for Authors
I have no further comments.
Comments on the Quality of English LanguageSee above